# An Explainable Artificial Intelligence Approach for Remaining Useful Life Prediction

Genane Youness [1,2,*] and Adam Aalah [3]

1   Laboratoire LINEACT CESI, IDFC, 92000 Nanterre, France
2   Laboratoire Cedric-MSDMA, 75003 Paris, France
3   Institut Polytechnique de Paris, 91120 Palaiseau, France; aalah.adam@gmail.com
*   Correspondence: gyouness@cesi.fr or genane.youness@lecnam.net

**Abstract:** Prognosis and health management depend on sufficient prior knowledge of the degradation process of critical components to predict the remaining useful life. This task is composed of two phases: learning and prediction. The first phase uses the available information to learn the system's behavior. The second phase predicts future behavior based on the available information of the system and estimates its remaining lifetime. Deep learning approaches achieve good prognostic performance but usually suffer from a high computational load and a lack of interpretability. Complex feature extraction models do not solve this problem, as they lose information in the learning phase and thus have a poor prognosis for the remaining lifetime. A new prepossessing approach is used with feature clustering to address this issue. It allows for restructuring the data into homogeneous groups strongly related to each other using a simple architecture of the LSTM model. It is advantageous in terms of learning time and the possibility of using limited computational capabilities. Then, we focus on the interpretability of deep learning prognosis using Explainable AI to achieve interpretable RUL prediction. The proposed approach offers model improvement and enhanced interpretability, enabling a better understanding of feature contributions. Experimental results on the available NASA C-MAPSS dataset show the performance of the proposed model compared to other common methods.

**Keywords:** prognostic and health management; remaining useful life; feature clustering; Explainable Artificial Intelligence (XAI)





## 1. Introduction

In recent years, the manufacturing sectors have taken a new approach, involving technological innovations, aimed to modernize and optimize production. This new concept is characterized by automizing mechanical processes using artificial intelligence. These techniques allow what is called "Industry 4.0" for the evolution of maintenance, and, more particularly, predictive maintenance. Due to artificial intelligence, predictive maintenance can anticipate anomalies (e.g., the life of computer servers, and the electrical installation of dysfunctional subway trains). In addition, intelligent maintenance requires a multistep approach, from monitoring, through analysis and decision support, to validation and verification. These steps are part of a discipline called PHM (prognostics and health management), linking the study of failure mechanisms and life cycle management.

Prognostics, in general, require two types of techniques: (1) an application-dependent technique that aims to detect precursors to estimate the system's health state, and (2) a prediction technique to predict the remaining useful life (RUL). Prognostic and health management (PHM) is an engineering field whose goal is to provide users with a thorough analysis of the health condition of a machine and its components [1]. Relying on human operators to manage atypical events is quite difficult when dealing with complex equipment and attempting to retrieve information and patterns of different equipment failures.

It aims to better manage the health status of physical systems while reducing operation and maintenance costs [2]. The implementation of PHM solutions is becoming increasingly important, and the prognostic process is now considered one of the main levers of action in the quest for overall performance [3]. It is generally described as a combination of seven layers that can be divided into three phases: observation, analysis, and action.

The remaining useful life (RUL) is the length of time a machine is likely to operate before it requires repair or replacement. By taking RUL into account, maintenance can be scheduled to optimize operating efficiency and avoid unplanned downtime. For this reason, estimating RUL is a top priority in predictive maintenance programs. Based on aging models (degradation of the monitored system), prognostics determine the health status of a system and predict the RUL. This prediction can be obtained using a physical-model-based approach [4], a data-driven approach [5], and a new hybrid approach [6] merging the first two approaches. Data-driven algorithms can be divided into three categories: statistical methods including Markov models, Weibull distribution, and Kalman filters; machine-learning-based methods such as support vector machines (SVMs) [7] and deep belief networks (DBNs) [8]; and deep learning methods [9]. Different works have shown the effectiveness of data-driven approaches [10], and especially deep learning, in estimating the RUL of a turbofan engine. A turbofan engine consists of several key components including a fan, compressor, combustion chamber, turbine, exhaust nozzle, and bypass duct, as well as various sensors to understand the condition of the engine. One of the advantages of data-driven deep learning approaches over the physics-based approach is that they can take into account the complexity of the structure and variety of sensors, which can affect the RUL of the turbofan engine, including operating conditions, maintenance history, and design features of the engine. To predict the RUL of a turbofan engine, several deep learning algorithms have been used, such as multilayer perceptrons, long short-term memories (LSTMs) [11], bidirectional LSTMs (Bi-LSTMs) [12], and convolutional neural network (CNN) algorithms [13], as well as combined algorithms to achieve higher accuracy. Advanced data-driven deep learning methods can have high predictive accuracy but are considered "black box" models because they suffer from a lack of interpretability. While these models can have high predictive accuracy, it can be difficult to understand how they arrive at their predictions, which can make it difficult to use the predictions to inform maintenance decisions. Explainable Artificial Intelligence (XAI) techniques such as SHapley Additive exPlanations (SHAP) values can be applied to explain the prediction process by providing information on how to estimate the contribution of each feature to the final prediction. However, their use in the predictive maintenance process has some limitations [14]. SHAP values assume that each feature contributes to the prediction independently of the others and is not able to capture their interactions, which can limit their accuracy and usefulness. Indeed, features interact with each other in ways that can be challenging and can affect the final prediction. Similar to many other permutation-based interpretation methods, SHAP values methods suffer from including unrealistic data instances when features are correlated [15]. They are also subject to limited human interpretation. These issues have received less attention in the scientific literature. To address these limitations, a deep-learning-based feature clustering with SHAP values approach is proposed to offer an efficient and interpretable model, leading to more accurate RUL predictions.

This study aims to answer questions about how to improve preprocessing to help with both interpretability and model performance as well as how to use predictions to build confidence in predictive maintenance models. The goal is to highlight the benefits of Explainable AI in the prognostic lifetime estimation of turbofan engines.

The accuracy of the model's prediction depends on the quality and relevance of the data that influence the engine's RUL. It is difficult to build accurate models with noisy data or complex features and to identify patterns. To address the challenge, a number of techniques can be used, such as dimensionality reduction, which reduces the number of dimensions. Unfortunately, popular techniques such as principal component analysis

(PCA) would generally transform the data in a way that damages the interpretability of the model [16]. This is because each transformed feature is a combination of all of the original features. Feature clustering tools capture relations between features that can preserve predictive power and yet retain interpretability.

The main contributions of this paper are summarized below:

- We propose a deep-learning-based feature clustering with SHAP values to explain the model's RUL estimation. Feature clustering is used to capture the relationship between features that can potentially influence the final prediction and improve interpretability, enabling a better understanding of feature contributions. It could be considered part of the Explainable AI field, as it promotes a better comprehension of the model. Using the feature clustering eliminates the need to use complex models; instead, an LSTM model with a single hidden layer is used. In the post-model phase, SHAP values are applied to facilitate the interpretation of RUL prediction and help determine the cause of the engine's failure.
- We discussed and analyzed the settings and effects of some critical parameters as well as the comparison with other methods. Therefore, we open the code of this experiment, in order to contribute to future developments.

To present our work, this paper is organized as follows: Section 2 explains Explainable Artificial Intelligence's (XAI's) need in PHM and reviews related work in this field. Section 3 provides an overview of previous related work to our study. Section 4 outlines the techniques used in this study and the proposed framework. An overview of the dataset and experiments setting are illustrated in Section 5. Results are presented and discussed in Section 6. Section 7 concludes the article.

## 2. Explainable Artificial Intelligence's (XAI) Need in PHM

System prognostics is a safety-sensitive industry field. It is therefore crucial to ensure the use of a properly regulated AI in this area. Deep learning networks are artificial neural networks that include many hidden layers between the input and the output layers. These approaches often offer better performance but are accused of being black boxes. They suffer from a lack of interpretability and analysis of prediction results. It is difficult to know how information is processed in the hidden layer. Therefore, explanatory capabilities are needed in RUL prediction to improve system reliability and provide insight into the parts that caused the engine failure. Explainable AI (XAI) has been presented as a solution to this problem. It is able to analyze the black box inside, verify its processes, and provide an understandable explanation of the logic behind the prediction. Several approaches can be applied to Explainable AI systems for prediction [17].

Carvalho et al. [18] classified the interpretability methods into three groups according to the moment when these methods are applicable: before (pre-model), during (in-model), or after (post-model) the construction of the model to predict. Some authors [18,19], consider that PCA, distributed stochastic neighbor embedding (t-SNE), and clustering methods can be classified under pre-model interpretability and can be part of the XAI field. The field of in-model interpretability is mainly focused on intrinsically interpretable models that are specifically designed to be transparent, such as decision trees, rule-based models, Bayesian models, and hybrid models.

Post-model interpretability refers to the improvement of interpretability after a model has been built (post hoc). Two approaches can be model-specific or model-agnostic. Model-specific interpretability includes self-explanation as the additional output, such as attention mechanisms [20] and capsule network (CapsNets) [21] that focus on the most relevant parts of the input data when making a prediction. Other examples of model-specific interpretability techniques include saliency maps and layer-wise relevance propagation. The post hoc model-agnostic interpretability approach consists of analyzing root causes by manipulating inputs and visualizing outputs without relying on the trained model. These approaches include local model analysis, global model analysis, and feature importance analysis. These methods can be used to identify the input features that contribute to the

predictions and have the greatest impact on the output of the model. In this study, SHAP values [22] are used on synthetic features to identify their contribution to the prediction and find the key factors that affect the RUL of a turbofan engine.

## 3. Related Work

The C-MAPSS dataset is one of the most used datasets for the goal of improving RUL estimations. Whether using machine learning models or deep learning models, many approaches and methods have been developed and proposed throughout the years. Wang et al. [23] built a fusion model that extracts features from the data based on a broad learning system (BLS) and integrates LSTM to process time series information. Heimes [24] was the first to implement a recurrent neural network (RNN) for RUL prediction. de Oliveira da Costa et al. [25] proposed a domain adversarial neural network (DANN) approach to learning domain-invariant features using LSTM. Jiang et al. [26] used a fusion network combined with a bidirectional LSTM network to estimate the RUL with the use of sequenced data. Zhao et al. [12] used a bidirectional LSTM (BiLSTM) approach for RUL estimation by taking sequence data in bidirection, and a model optimization was necessary to obtain the best results possible. Zhang et al. [27] proposed an LSTM-fusion architecture, concatenating separate LSTM subnetworks, on sensor signals with feature window sizes. Listou Ellefsen et al. [28] proposed a semisupervised approach based on LSTM using a genetic algorithm (GA) to adjust the diverse amount of hyperparameters in the training procedure. Zheng et al. [29] used a deep LSTM model for RUL estimation feeding sequenced sensor data to the model to reveal hidden features when multiple operational conditions are present. Lee [30] adopted a new approach by normalizing the sensor data per operational condition, as well as rectifying the RUL; it was achieved by setting an early RUL, removing the initial cycles where the engine is at a healthy state and only focusing on the degradation period. Recently, Palazuelos et al. [31] extended capsule neural networks for fault prognostics, particularly remaining useful life estimation. Chen et al. [32] applied an attention-based deep learning framework hybrid LSTM with feature fusion able to learn the importance of features. Quin et al. [33] proposed a slow-varying dynamics-assisted temporal CapsNet (SD-TemCapsNet) to simultaneously learn the slow-varying dynamics and temporal dynamics from measurements for accurate RUL estimation. Li et al. [34] proposed a cycle-consistent learning scheme to obtain a new representation space, considering variations in the degradation patterns of different entities. Ren et al. [35] proposed a lightweight and adaptive knowledge distillation for enhancing industrial prediction accuracy. To address the sensor malfunction problem, the authors in [36] introduced adversarial learning to extract generalized sensor-invariant features. To handle the presence of uncertainties and find the key features that contribute to the precipitation forecasting process. Manna et al. [37] incorporated a rough ensemble on fuzzy approximation space (RSFAS) into the LSTM method.

## 4. Materials and Methods

### 4.1. Feature Clustering: ClustOfVar

As mentioned previously, feature clustering can be classified as pre-model interpretability and is part of the XAI domain. It facilitates visualization and understanding of relationships between features and pattern identification. Feature clustering obtains clusters of related and redundant features. It creates synthetic features that best summarize the relevant information provided by the initial features. This approach can be considered as a dimension-reduction step and seems to be a good alternative to principal component analysis. Indeed, it allows for removing the redundancies of information in the available p features. The synthetic features will be constructed only with the features of the group, contrary to the principal component analysis where the synthetic features are constructed with all the features. It thus makes it easier to interpret the synthetic features. The homogeneity criterion of a cluster is the squared correlations summarizing the features in the cluster. The ascendant hierarchical clustering method is used. At each stage, the two clusters that

minimize the loss of homogeneity are aggregated. Each feature is then reassigned to the cluster that is closest to it, i.e., the one with the highest squared correlation between the feature and the cluster representative. To evaluate the stability of the partition, a bootstrap approach is used; it allows the determination of suitable numbers of clusters. ClustOfVar approaches [38] implemented in the R packages are used.

### 4.1.1. Definition of the Synthetic Feature of the Cluster

Each cluster $C_k$ can be summarized by a numerical synthetic feature noted $y_k$ defined as follows:

$$y_k = \arg \max_{u \in R^n} \sum_{j \in C_k} r^2(x_j, u) \tag{1}$$

$r^2(x_j, u) \in [0, 1]$ is the square of the linear Pearson correlation between the feature $x_j$ and the synthetic feature $u$.

### 4.1.2. Homogeneity $H$ of the Cluster $C_k$

The homogeneity $H$ of the partition $P_k = (C_1, \ldots, C_k)$ is defined as the sum of the homogeneities of its $k$ clusters:

$$\mathcal{H}(P_K) = \sum_{k=1}^{K} H(C_k) \tag{2}$$

The goal of the hierarchical ascending classification algorithm is to find a partition of a set of p features that maximize the homogeneity $H(P_k)$.

### 4.1.3. The Hierarchical Clustering Algorithm

The aim is to find a partition that maximizes the homogeneity function $H$ defined in (2) in such a way that the features within a cluster are strongly related to each other. The hierarchical bottom-up classification algorithm starts with the partition into p clusters. At each step, the two classes $A$ and $B$ that minimize the loss of homogeneity are aggregated. Thus, the two clusters with the smallest dissimilarity $d$ are chosen. It is defined as follows:

$$d(A, B) = H(A) + H(B) - H(A \cup B) \tag{3}$$

Using this aggregation measure, the new partition maximizes $H$ among all partitions obtained by aggregating two clusters.

### 4.1.4. Stability of Feature Partitions

To assess the stability of all partitions into 2 to p − 1 classes resulting from the hierarchical clustering, a bootstrap resampling procedure is used to determine the appropriate number of feature classes that can be retained for analysis. The right choice for this number of classes is the one that retains the most stable partition. B bootstrap replications of the original data are generated and the associated B dendrograms are obtained. The partitions of these B dendrograms are compared to the partitions of the original hierarchy through the adjusted Rand index [39]. The stability of the partition is evaluated based on the average of the values of the adjusted B Rand indices. The closer the Rand index value is to 1, the more similar the partitions are.

### 4.2. Long Short-Term Memory Neural Network

A type of recurrent neural network was developed by [40], long short-term memory (LSTM), becoming a powerful model for processing temporal data with complex structures. It was a solution to the problem of the optimal gradient of the backpropagation to change the weights of the network. This is a promising approach to service life prediction, providing valuable short- and long-term information. The LSTM network is shown in Figure 1 for a unit at step t with no indication of weight.

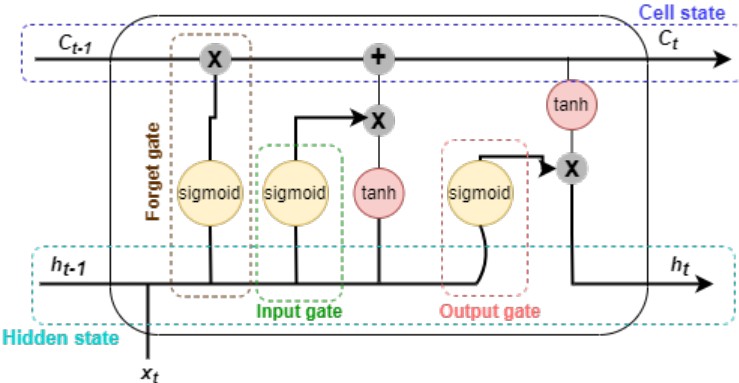

**Figure 1.** Structure of an LSTM network unit.

Each unit is linked to a hidden state $h$ and to a state $c$ of the cell that serves as a memory. Three gates regulating the flow of information represent the structure of the LSTM network: the input gate allows or blocks the update of the cell $c$, the forget gate allows the reset of the cell state, and the output gate controls the communication from the cell $c$ to the output of the unit. The activation function used is the hyperbolic tangent function. It transforms the values between $-1$ and $1$ and prevents the outputs from becoming large.

$$
\begin{aligned}
F_t &= \sigma(W_F x_t + U_F h_{t-1} + b_F) \quad \text{(forget gate)} \\
I_t &= \sigma(W_I x_t + U_I h_{t-1} + b_I) \quad \text{(input gate)} \\
O_t &= \sigma(W_O x_t + U_O h_{t-1} + b_O) \quad \text{(output gate)} \\
c_t &= F_t \circ c_{t-1} + I_t \circ \tanh(W_c x_t + U_c h_{t-1} + b_c) \\
h_t &= O_t \circ \tanh(c_t) \\
o_t &= f(W_o h_t + b_o)
\end{aligned}
\tag{4}
$$

The inputs of each gate are weighted by the gate weights and by a bias. There are four weight matrices whose dimensions depend on the dimensions of $h_{t-1}$ and $x_t$:

- $W_F$: weight of the forget gate;
- $W_I$: weights of the input gate;
- $W_c$: weights of the data that will be combined with the input gate to update the cell state;
- $W_o$: weights of the output gate.

In our proposed model, we used one LSTM layer to extract the temporal synthetic features included in the previous time windows with size $T_w$ before estimating the RUL.

### 4.3. SHAP (SHapley Additive Explanations)

SHAP (SHapley Additive Explanations), proposed in [22], is a method used to improve the interpretability of artificial intelligence methods. Issued from game theory, it computes Shapley values [41] to each predictor to indicate its contribution to the final result. The idea is to average the impact of a feature over all possible combinations of features in a linear model. The Shapley value $\phi_j$ is defined as follows:

$$
\phi_j(f, x) = \sum_{S \subseteq \{x_1, \dots, x_p\} \setminus \{x_j\}} \frac{|S|!(p - |S| - 1)!}{p!} (f_x(S \cup \{j\}) - f_x(S))
\tag{5}
$$

where $p$ is the number of all features, $S$ is a subset of features used in the model, and $j$ is the $j$th feature. $f_x$ is the prediction function of a single input $x$, defined as

$$
f_x(S) = \int \hat{f}(x_1, \dots, x_p) dP_{x \notin S} - E_X(\hat{f}(X))
\tag{6}
$$

It performs several integrations for each feature that does not contain $S$. A linear $g$ model is finally fitted to the features and their impacts, represented as follows:

$$f(x) = g(x') = \phi_0 + \sum_{i=1}^{p} \phi_i x_i \tag{7}$$

where $f(x)$ represents the original model, $g(x)$ is the explanation model, $x'$—simplified input, such that $x = h_x(x')$, it has several omitted features, and $\phi_0 = f(h_x(0))$; it represents the model output with all simplified inputs missing. Shapley values are the only set of values that satisfy the following three properties: local accuracy, missingness, and consistency. Baptista et al. [42] found that SHAP values are very close to three properties: monotonic, trendable, and prognosable, as model complexity increases. David Solís et al. [43] introduced a new proxy, "acumen", which aims to detect whether the XAI method depends on temporal dependencies. Their results showed that SHAP provided the best "acumen".

### 4.4. Proposed Method

The LSTM-based feature clustering with SHAP values approach is shown in Figure 2. The C-MAPSS dataset FD004 of NASA is selected, and is simulated under six operating conditions and two failure modes. The data are first normalized based on operating conditions for the training and test sets. Exponential smoothing is then applied to reduce noise. A fixed-length sliding time window (TW) and RUL rectification are used to emphasize the degradation part when training the model. The data are then processed using a feature clustering technique to reduce the dimensionality of the input space and enhanced interpretability. It represents the new dataset to predict RUL. The synthetic feature training data are trained using an LSTM model. The model learns using one hidden layer. Once the training is complete, the final model is set, and the prediction and the RUL error are then calculated by applying the clustered test data to this model. Finally, among the Explainable Artificial Intelligence techniques, SHAP values are applied to clustered features. It provides a way to understand the role of each factor cluster in an aircraft's turbofan engine. This allows us to determine the source of degradation at the engine level in order to facilitate maintenance.

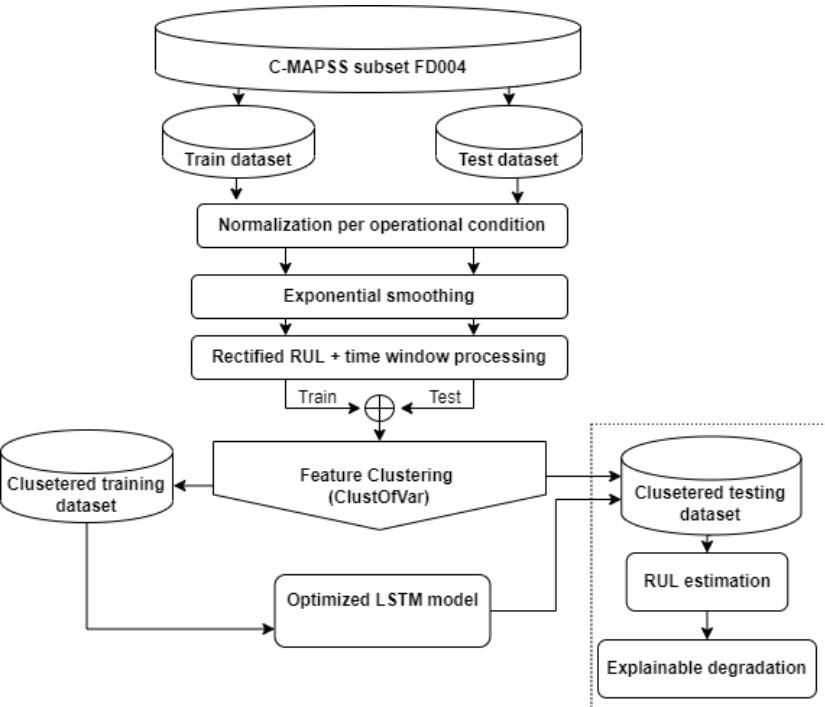

**Figure 2.** The flowchart of the proposed method.

## 5. Experiment Settings

In this section, we explain the proposed model to predict the RUL. The experimental environment was DELL-Intel Core vPRO i9, 64 GB RAM, 16 GB GPU NVIDIA RTX A5000, and Microsoft 10 operating system. The models and algorithms were implemented in Python language, with an exception when using the R language for the ClustOfVar package to compute the feature clustering. The frameworks used were Tensor-Flow and Keras on Jupyter-Notebooks IDE. The experiment's code is available on GitHub (https://github.com/adam-aalah/Feature-clustering-and-XAI-for-RUL-estimation, (accessed on 17 May 2023)).

### 5.1. Data Overview

NASA has developed a generic military turbofan engine simulation [44], the Modular Aero-Propulsion System Simulation (C-MAPSS), which has been released for public use to rapidly implement and test control algorithms on a validated engine model. C-MAPSS simulates an engine model of the 90,000 lb thrust class and the package includes an atmospheric model capable of simulating operations at an altitude ranging from sea level to 40K ft, Mach number from 0 to 0.90, and sea-level temperatures from $-60$ to 100 °F.

The C-MAPSS engine contains five rotating components, Figure 3. These components can be divided into two subcategories: a low-pressure shaft that contains fan, low-pressure compressor (LPC), and a low-pressure turbine (LPT); and a high-pressure shaft that contains a high-pressure compressor (HPC) and a high-pressure turbine (HPT).

To comprehend how the engine operates, we need to understand each component and its role (Figure 3). The fan is the first component located at the entrance of the turbofan engine. The rotation of its blades causes the suction of large quantities of air. Followed by the compressor, being the first component in the engine core, it contains two subcomponents: a low-pressure compressor (LPC) and a high-pressure compressor (HPC). The compressed air obtained is then pumped into the combustion chamber, where the injected fuel is mixed and burned. This provides a high-temperature airflow that is supplied to the turbine. The turbine contains two subcomponents: a high-pressure turbine (HPT) and a low-pressure turbine (LPT). The high-energy airflow comes out of the combustor chamber into the turbine, causing the turbine's blades to rotate. The air sucked into the turbofan engine is split into two parts; the first part moves to the combustor chamber, passing through the compressor, and the second part moves through the bypass. The role of the bypass is to increase thrust without increasing fuel consumption, as well as to provide cooling air.

We choose to work with the C-MAPSS subset FD004, as it is the most complex dataset out of four subsets (see Table 1), containing two fault modes and six operational conditions which are detailed in Table 2. The data was collected using 21 sensors located in different parts of the engine presented in Figure 3. The different sensors are summerized in Table 3 as well as their description. The experiments can be conducted through three stages: preprocessing, training and RUL estimation, and then explaining the degradation.

**Table 1.** Detailed information of the C-MAPSS subsets.

|  | FD001 | FD002 | FD003 | FD004 |
|---|---|---|---|---|
| Units (Train) | 100 | 260 | 100 | 249 |
| Units (Test) | 100 | 259 | 100 | 248 |
| Operation condition | 1 | 6 | 1 | 6 |
| Fault modes | 1 | 1 | 2 | 2 |

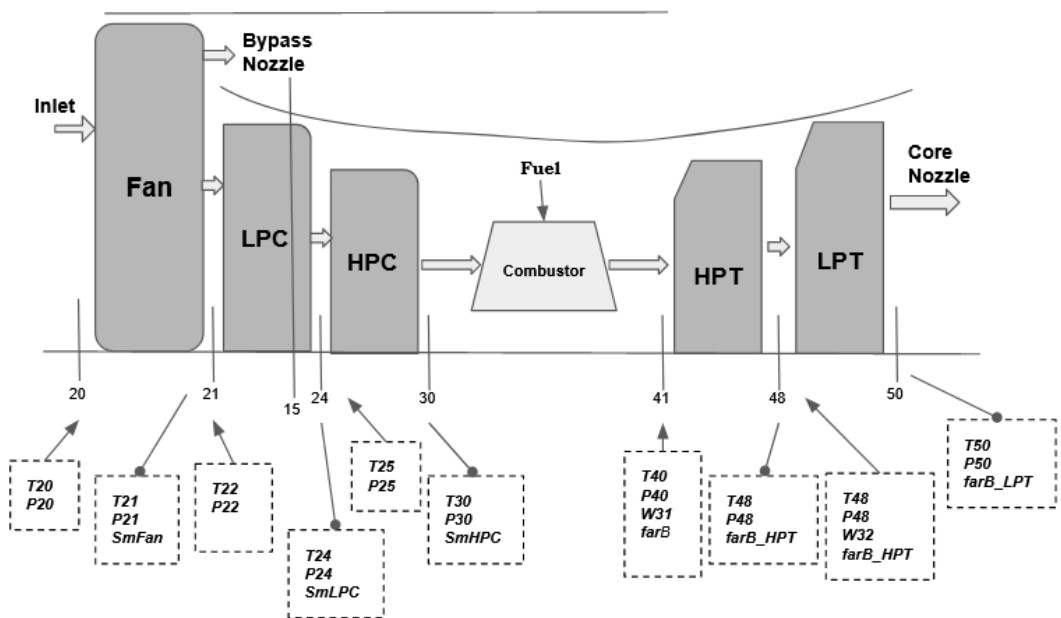

**Figure 3.** C-MAPSS engine components.

**Table 2.** Operational conditions at which the data were simulated.

| ID | Name | Alt, ft | Mach | TRA, deg | Fan Speed, rpm | Core Speed, rpm | Fuel Flow, pps | T48, °R |
|----|------|---------|------|----------|----------------|-----------------|----------------|---------|
| 1 | FC09 | 42K | 0.84 | 100 | 2212 | 8317 | 1.518 | 1744 |
| 2 | FC08 | 35K | 0.84 | 100 | 2223 | 8346 | 2.120 | 1750 |
| 3 | FC07 | 25K | 0.62 | 60 | 1915 | 8006 | 1.670 | 1534 |
| 4 | FC06 | 20K | 0.70 | 100 | 2324 | 8719 | 3.863 | 1909 |
| 5 | FC05 | 10K | 0.25 | 100 | 2319 | 8774 | 4.661 | 1947 |
| 6 | FC01 | 0K | 0 | 100 | 2388 | 9051 | 6.835 | 2072 |

**Table 3.** C-MAPSS 21 sensor readings.

| Symbol | Description | Unit |
|--------|-------------|------|
| T20 | Total temperature at fan inlet | °R |
| T24 | Total temperature at LPC outlet | °R |
| T30 | Total temperature at HPC outlet | °R |
| T50 | Total temperature at LPT outlet | °R |
| P20 | Pressure at fan inlet | psia |
| P15 | Total bypass-duct | psia |
| P30 | Total pressure at HPC outlet | psia |
| Nf | Physical fan speed | rpm |
| Nc | Physical core speed | rpm |
| epr | Engine pressure ratio (P50/P20) | - |
| Ps30 | Static pressure at HPC outlet | psia |
| $\phi$ | Ratio of fuel flow to Ps30 | pps/psia |
| NRf | Corrected fan speed | rpm |
| NRc | Corrected core speed | rpm |
| BPR | Bypass ratio | - |
| farB | Burner fuel–air ratio | - |
| htBleed | Bleed enthalpy | - |
| $Nf_{dmd}$ | Demanded fan speed | rpm |
| $PCNfR_{dmd}$ | Demanded corrected fan speed | rpm |
| W31 | HPT coolant bleed | lbm/s |
| W32 | LPT coolant bleed | lbm/s |

*5.2. Preprocessing Approach*

5.2.1. Normalization According to Operation Condition

Since FD004 was simulated under six different operational conditions [30], normalization was then performed to each operational condition. The operational condition is identified by the Mach number, altitude, and ambient temperature of an aircraft engine. It is a controlled setting or environmental parameters. Normalizing the sensors to this information has a significant impact on the performance of the model. Normalized values are found within the range $[-1, 1]$ using the min–max normalization method:

$$\text{norm}\left(x_t^{i,j}\right) = \frac{x_t^{i,j} - \min\left(x^j\right)}{\max\left(x^j\right) - \min\left(x^j\right)} - 1 \tag{8}$$

5.2.2. Data Smoothing

Normalizing the data is not enough to produce a reasonable and accurate RUL estimation due to the large amount of noise present in the data. Exponential smoothing is applied. It uses a parameter $\alpha$ that controls the smoothing factor. The value of $\alpha$ lies between 0 and 1. The fitted values are written as follows:

$$\hat{y}_{t+1|t} = \alpha y_t + (1 - \alpha)\hat{y}_{t|t-1} \tag{9}$$

- $\alpha = 0$: signifies that those future forecasted values are the average of historical data, giving more weight to historical data.
- $\alpha = 1$: signifies that future forecast values are the results of the recent observation, giving more weight to recent observations.

A value close to 1 indicates fast learning (that is, only the most recent values influence the forecasts), whereas a value close to 0 indicates slow learning (past observations have a large influence on forecasts). A grid search was performed to find the best value of $\alpha$ in the range of [0.1; 0.9], which turned out to be equal to 0.3. However, the effect of the smoothing value $\alpha$ on the performance of the model will be investigated in further experiments.

5.2.3. Feature Clustering

Feature clustering is illustrated on the combined preprocessed train and test sets, and then we carried out a hierarchical feature clustering, reducing the inputs of the LSTM neural network. The stability criterion is employed based on a bootstrap approach to choose the number of clusters using the "ClustOfVar" R-package [38].

The stability in Figure 4 is achieved up to seven clusters and then restabilizes from nine clusters. Based on the stability, the choice of three clusters gives a gain in cohesion equal to 66.98%, while the choice of seven clusters gives a gain in cohesion equal to 90.09%. Therefore, it seems reasonable to keep seven clusters, as it allows us to give a deeper insight into how the engine's degradation occurs.

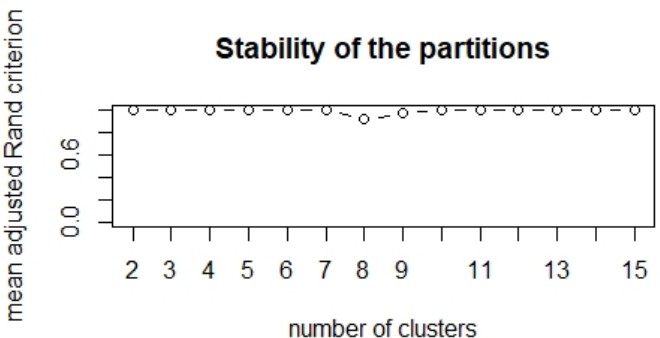

**Figure 4.** Stability criterion for the partitions obtained by the hierarchical clustering of the features.

According to Figure 5, the sensors are distributed across the seven clusters. These synthetic features are structured as follows:

$$\text{cluster 1} \quad \begin{aligned} &= 7.856395 - 3.887295 \times T24 - 3.847008 \times T30 - 3.245412 \times T50 \\ &\quad - 2.964724 \times Ps30 - 3.533685 \times htBleed \end{aligned} \tag{10}$$

$$\text{cluster 2} = -3.447993 + 4.611694 \times P15 \tag{11}$$

$$\text{cluster 3} = -3.345230 + 5.308207 \times P30 + 5.164441 \times \phi \tag{12}$$

$$\text{cluster 4} = -4.890251 + 6.789197 \times Nf + 6.588968 \times NRf \tag{13}$$

$$\text{cluster 5} = -3.642547 + 7.101891 \times Nc + 7.607409 \times NRc \tag{14}$$

$$\text{cluster 6} = -2.868859 + 8.863718 \times epr \tag{15}$$

$$\text{cluster 7} = -2.891498 - 4.090116 \times BPR + 5.704820 \times W31 + 5.630470 \times W32 \tag{16}$$

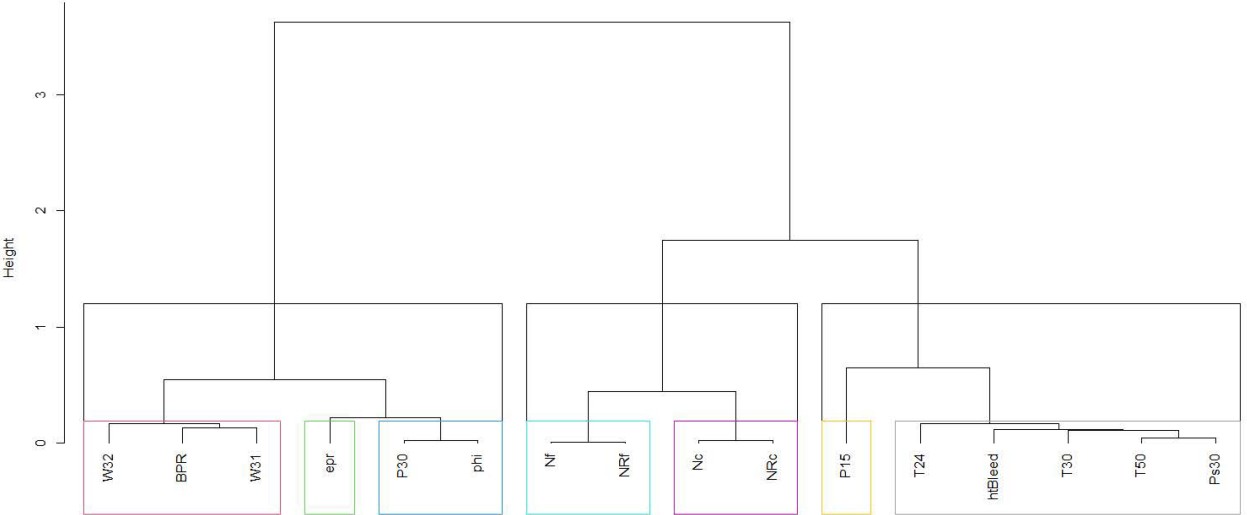

**Figure 5.** The dendrogram obtained by the hierarchical clustering of the features.

The synthetic feature scores for each observation are extracted. This represents the new dataset applied to predict the RUL.

### 5.2.4. Time Window and Rectified RUL

After reducing the noise using exponential smoothing, a fixed-length sliding time window (TW) processing is applied to convert multivariate time series data into sequential measurement cycle data. A time window is moved each time by one measurement cycle to generate a new TW. A longer length of TW can contain more local information and therefore impacts the performance of the proposed model. Simultaneously, it slows down the learning speed of the model. A grid search was performed on time window size for values $\in [30,60]$, using the LSTM model, and led to a TW value equal to 30. The engine fails at an unknown time and is healthy before the failure, so a linear degradation model is essential for model convergence. After fixing the time window to 30, and based on different values of rectified RUL [29], we set an early RUL value equal to 130. The engine is considered healthy until it reaches cycle 130, where degradation can be observed. Since all engines are healthy from cycle 130 onward, the remaining life of the train and test set

was rectified. This allowed the model to learn from the critical cycle interval instead of the whole life of the engine.

We examine the effect of the time window size and the rectified true RUL setting on the model performance in the subsequent experiments.

*5.3. Model Setup*

5.3.1. Hyperparameters

The proposed model is built with a simple long short-term memory neural network, LSTM. Its architecture is composed of one LSTM layer with an activation 'tanh', one dense layer with an activation function 'ReLu', followed by a dropout layer with an activation layer 'ReLu' and a dense (1) layer. The grid search cross-validation method is used for parameter optimization. The best hyperparameters with the lowest RMSE used in this study are shown in Table 4.

**Table 4.** Hyperparameters used for the LSTM model.

| Hyperparameter | Value |
|:---:|:---:|
| Number of Hidden Layers | 1 |
| Nodes | 128 |
| Dropout | 0.2 |
| Batch-size | 384 |
| Epochs | 20 |
| Learning-Rate | 0.01 |

Experiments were conducted on seven synthetic features with a time window equal to 30 and an early RUL value equal to 130. The Adam optimizer was applied, given the best results with a learning rate set to 0.01. For the proposed model, the LSTM used a single hidden layer with 128 nodes, and the dropout was set to 0.2.

5.3.2. Prognostic Metrics

To evaluate the model's performance, two metrics are used in this paper: root mean square error (RMSE) and the scoring function.

- RMSE evaluates the prediction error. It is represented as follows:

$$\text{RMSE} = \sqrt{\frac{1}{n}\Sigma_{i=1}^{n}(d_i)^2} \tag{17}$$

with $d_i = \text{RUL}_i' - \text{RUL}_i$

- Based on the PHM data challenge in 2008 [45], a scoring function, S-score, was defined to penalize a lot more late predictions in comparison to early predictions of the remaining useful life. It is described as follows:

$$\text{S-score} = \begin{cases} \sum_{i=1}^{n}\left(e^{-\frac{d_i}{13}} - 1\right) & \text{for } d_i < 0 \\ \sum_{i=1}^{n}\left(e^{\frac{d_i}{10}} - 1\right) & \text{for } d_i \geq 0 \end{cases} \tag{18}$$

where $n$ denotes the total number of test sets, the $\text{RUL}_i'$ represents the predicted RUL, the $RUL_i$ denotes the actual RUL, and $d_i$ represents the prediction error.

## 6. Results Analysis

To better estimate the remaining useful life of the units, we study the effect of different parameters on the model's performance, such as data smoothing parameter $\alpha$, the time window size, and the piecewise rectified RUL. The proposed model was used on our new preprocessed dataset containing synthetic features. Experimental results were

averaged by 30 independent trials to ensure the accuracy of the results and reduce the effect of randomness.

### 6.1. Parameters Study

We first adjusted the data smoothing parameter $\alpha$ to 0.2, 0.3, 0.7, and 0.9, consistent with previous studies. After 30 independent trials, the results are shown in Figure 6. The 95% confidence interval of the RMSE is displayed for each $\alpha$ value. The red dot indicates the average RMSE. The best value of the smoothing parameter $\alpha$ is equal to 0.3, which leads to the smallest root mean square error (RMSE).

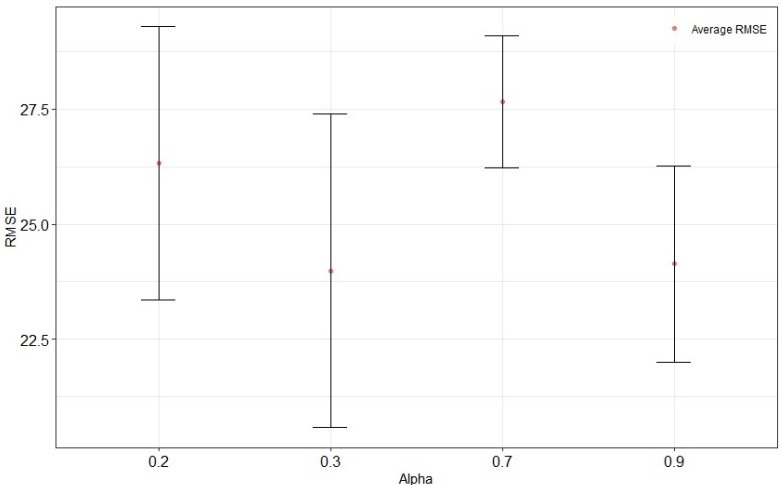

**Figure 6.** 95% CI for average RMSE by smoothing parameters $\alpha$.

Regarding the size of the time window, most studies using the C-MAPSS FD004 database set it at 30. Nevertheless, we believe that this parameter should be investigated. Setting $\alpha$ to 0.3, we adjusted the time window to 10, 20, 30, 40, and 50. Each configuration was repeated 30 times independently. The results in Figure 7 show that the RMSE first decreases as the window size increases and exceeds the value of 30, indicating that a window size that is too large does not help improve performance. This is because the current RUL is correlated with data from the most recent period, and the correlation decreases as the time interval increases. Including these data may result in too much noise. On the other side, the test results' variance increases with the time window's size, indicating that the model learning is indeed perturbed by noise. The best results are obtained with a time window size of 30.

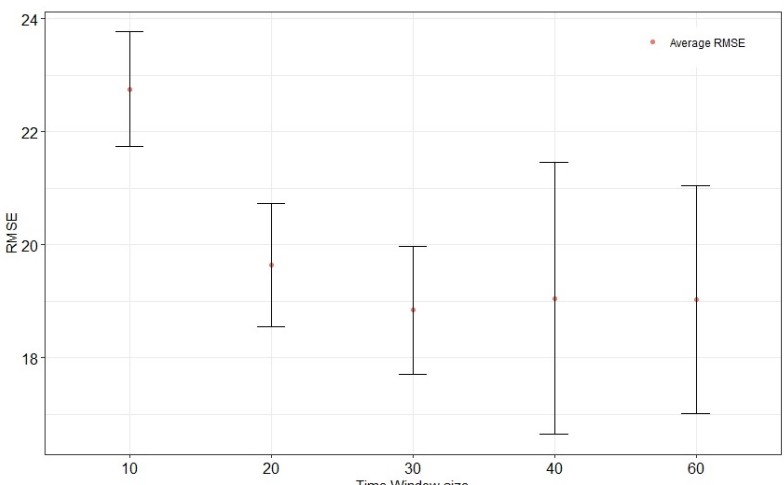

**Figure 7.** 95% CI for average RMSE by time window sizes.

In addition, we studied the impact of the rectified RUL, as per the window size experiments. Setting a time window to 30, we adjusted RUL early to 130, 140, and 150 and repeated each trial 30 times independently. The results in Figure 8 show that the lowest RMSE is achieved with a rectified RUL equal to 130.

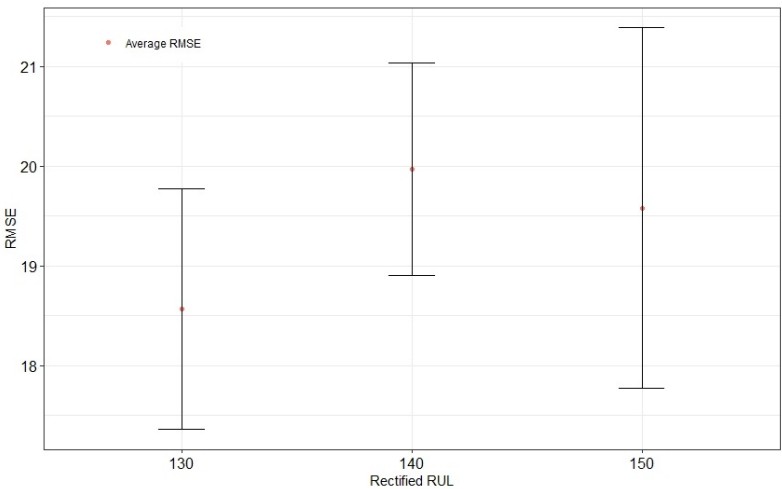

**Figure 8.** 95% CI for average RMSE by rectified RUL values.

*6.2. Ablation Study*

In this section, we perform an ablation study to show the effectiveness of using feature clustering in preprocessing, improving model performance and overfitting problems by reducing the influence of input sensors. First, our results are compared to those obtained by deploying a one, two, and three-hidden-layer LSTM network without dimension reduction, presented in Figure 9. Second, the results are compared to those obtained by the PCA dimension reduction method with 3 PCs, 4 PCs, and 16 PCs using a single hidden layer LSTM network, found in Figure 10. This comparison mainly shows the motivation behind using feature clustering. This approach extracts information from correlated features to reveal hidden patterns, thus providing a simpler model with fewer hidden layers. Our subsequent experiments validate this assumption.

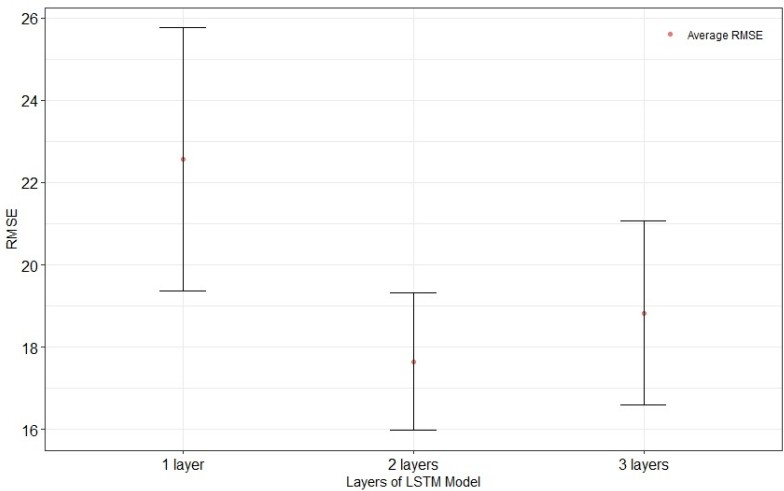

**Figure 9.** Impact of LSTM layers using all sensors in terms of RMSE.

The sensors were smoothed using $\alpha$ equal to 0.3, the time window equal to 30, and rectified RUL set to 130. The results in Table 5 and Figure 9 show that when no dimension reduction is performed, two layers are required to achieve better performance with the LSTM network. Furthermore, the experimental results in Figure 11 show also that increas-

ing the complexity of the LSTM network does not lead to the prediction accuracy found with feature clustering.

**Table 5.** Results using different inputs: All sensors, feature clustering, and PCA components.

| Method | Input Features + Model | RMSE | S-Score |
|---|---|---|---|
| All sensors + LSTM | 1 layer | 22.56 | 5127.79 |
| | 2 layers | 17.64 | 1823.05 |
| | 3 layers | 18.83 | 3063.82 |
| PCA + LSTM (1 layer) | 16 PC * | 20.10 | 2949.82 |
| | 4 PC * | 19.84 | 2524.4 |
| | 3 PC * | 20.63 | 2924.69 |
| Proposed method | 7 clusters | 16.14 | 1299.19 |

* Principal components.

According to Figure 10, the best results using the PCA method were obtained with four principal components catching all the variance in the original data. This result indicates that reducing the number of components reduces the complexity of the model, which improves the prediction accuracy. However, with excessive dimension reduction, the features required for the prediction cannot be learned. In addition, our model performed better than the model using PCA dimension reduction (Figure 11).

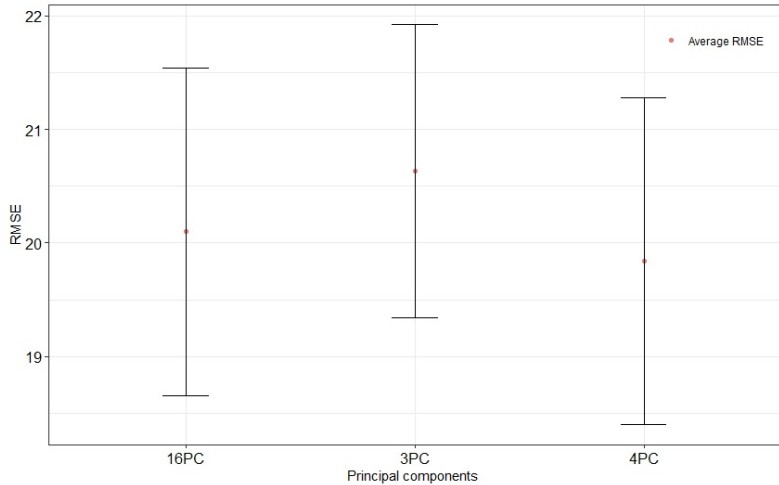

**Figure 10.** Impact of PCA dimension reduction with LSTM (1 layer) in terms of RMSE.

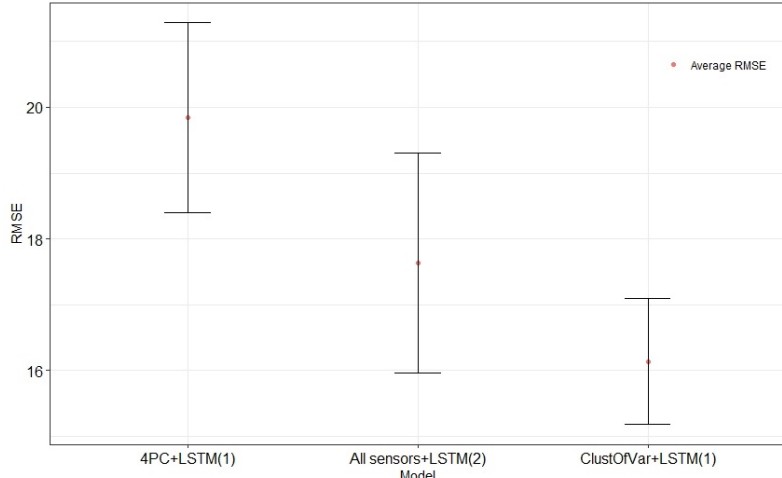

**Figure 11.** Comparative study.

Compared to the other models, in Table 5, the proposed model performs better, with the lowest average RMSE value (16.14) and the lowest average S-score (299.19). It also provides the smallest standard deviation (values in brackets), showing that the performance of our model is more accurate. These results indicate that, with feature clustering as a part of the preprocessing, reducing the complexity of the model does not decrease the information needed for learning.

### 6.3. Case Study

The estimated RUL values at each run are compared to the true RUL values for 4 different engines randomly selected, for the training subset of FD004, when using RUL. The aqua-clear blue, orange, and red lines correspond, respectively, to the estimated RUL values found by 2-layer all-sensor LSTM, 4PCA + LSTM (1 layer), and our proposed model with 7ClustOfVar + LSTM (1 layer).

The engines are sorted by ascending order of real RUL values and are represented in dark blue, demonstrating the lifetime of the aircraft engine until its failure. These real values are used to observe the performance of our model, indicating the error between the true and the predicted RUL.

The comparison between the estimated RUL using the different methods for randomly selected engines is shown in Figure 12. It highlights that the estimated RUL values with the proposed method are closer to the true RUL values of the training subset than to those estimated using the four principal components or the LSTM model with two hidden layers without performing any type of dimension reduction as inputs.

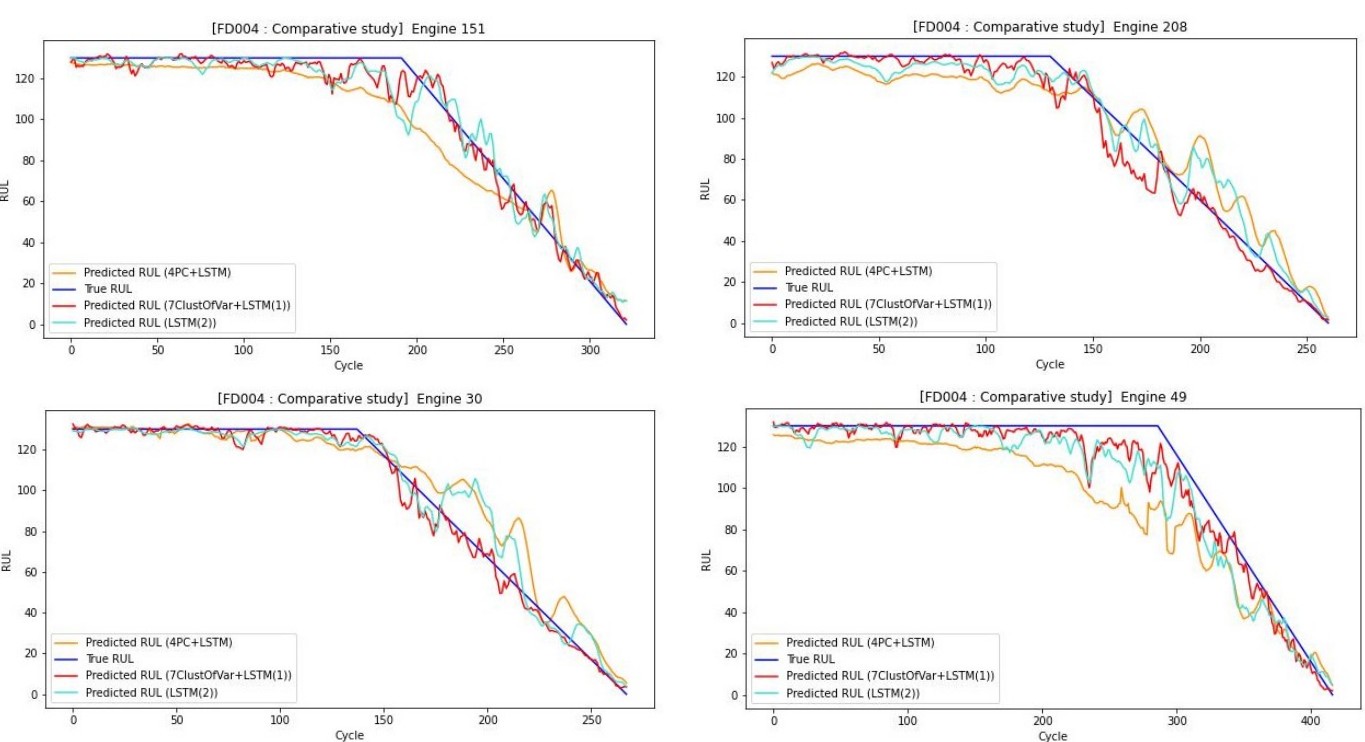

**Figure 12.** Comparing the estimated RUL using different methods 7ClustOfVar + LSTM(1), 4PC + LSTM(1), and LSTM(2), for randomly selected engines.

Figure 12 indicates that when using the clustered features as input, the model tends to learn the hidden pattern available within these clusters well, whereas when using either four principal components an LSTM model with two hidden layers without performing any type of dimension reduction, it fails to learn well, and either estimates the RUL early or late, which can be an issue when considering the turbofan engines, as it can lead to a disastrous outcome in a real-life scenario.

The prediction results of randomly selected engines from FD004 train sets are presented in Figure 13. The majority of the errors based on [46] fall within the confidence interval, defined by the lower bound $d_i = -13$ and the upper bound $d_i = 10$, proving that the model's performance was well trained.

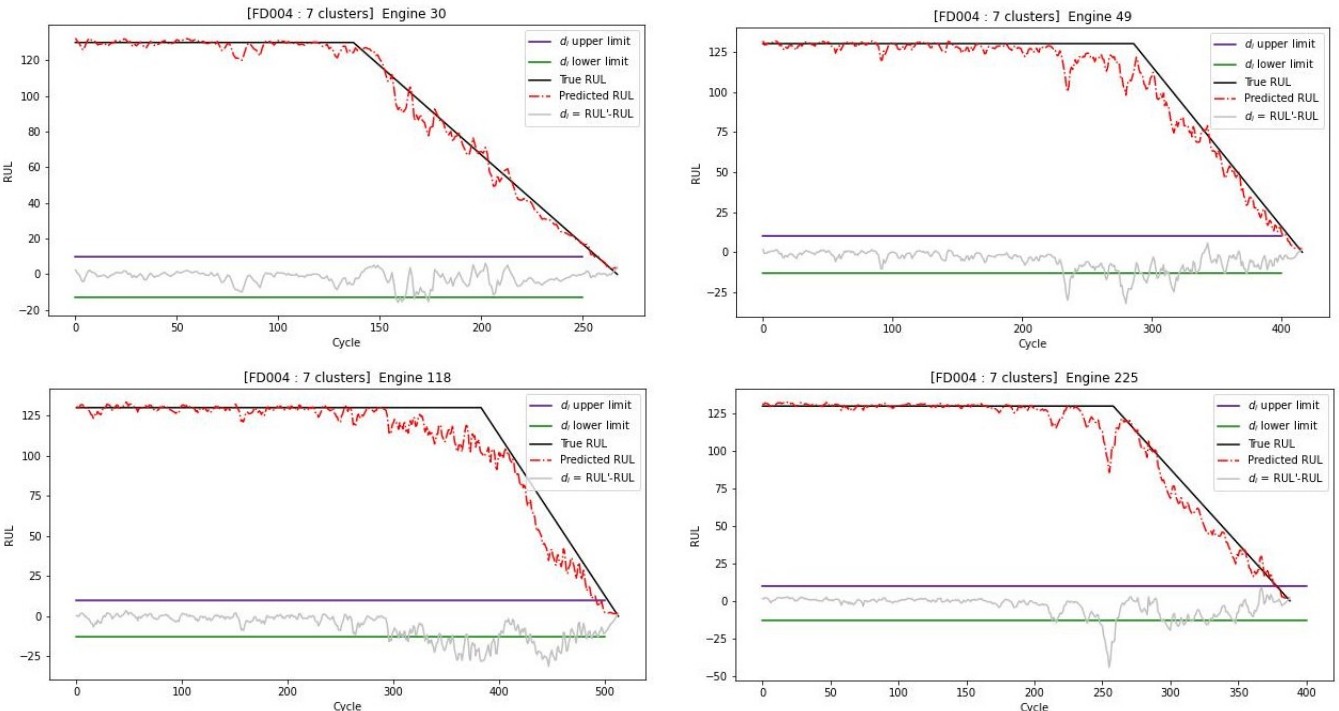

**Figure 13.** Estimated RUL using our proposed model for randomly selected engines.

In order to evaluate the model's performance, graphically, the predicted error $d_i$ for the test subset of FD004 is represented. The green and purple lines correspond, respectively, to the lower limit $d_i = -13$ and the upper limit $d_i = 10$.

As illustrated in Figure 14, most of the values fall within this confidence interval determined with the lower and upper limit of $d_i$, showing the efficiency of the proposed model's estimation. These values help identify the values that were correctly predicted within the confidence interval, with the blue color and those which were either an early or a late prediction, respectively with colors yellow and red. A comparison between the predicted RUL and the rectified true RUL can also be seen in Figure 15.

After examining the predicted error values, it is necessary to evaluate how the error values are translated in terms of actual values, which is why a comparison between predicted and true RUL values is necessary. The dashed red colored line in Figure 15 represents the predicted RUL values, and the dashed gray colored line represents the given ground truth RUL of the 248 engines of the test subset of FD004. Generally, the predicted values for each engine have a small difference from the true RUL, with some values that were predicted either early or late. Overall, our model performs well in estimating the remaining useful life of the engines, despite some engines being too noisy and affecting the model's training and performance, especially when the model predicted fewer late RUL values.

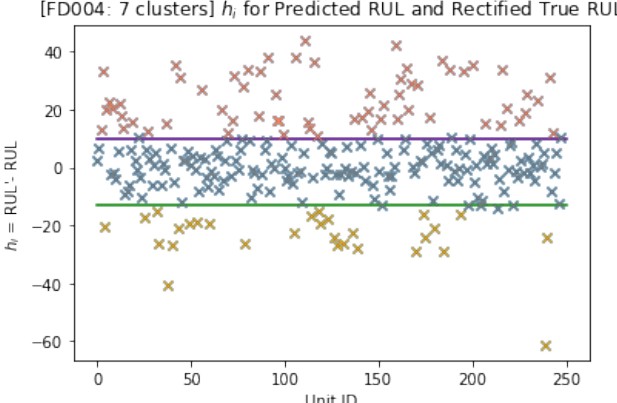

**Figure 14.** The Predicted error values for the test FD004 subset using the proposed model with respect to the confidence interval determined with the lower and upper limit of $d_i$.

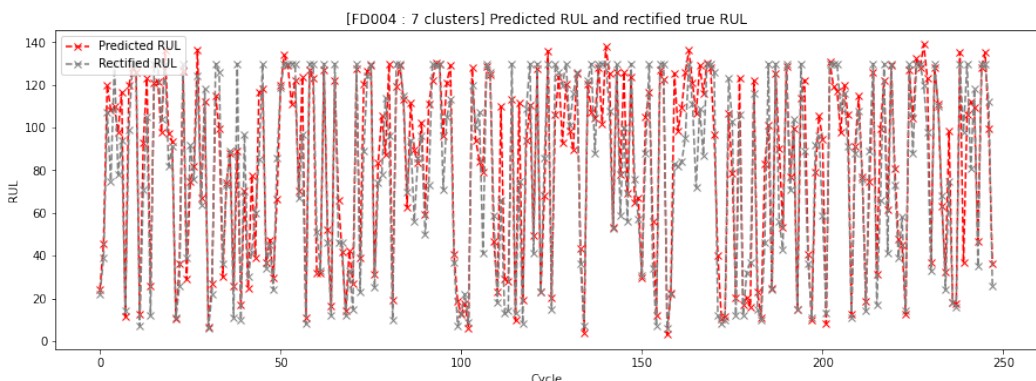

**Figure 15.** Comparison of the rectified true values and predicted values of RUL, of all engines in the test set using the proposed model.

*6.4. Comparison with Other Work*

A comparison of the proposed method with state-of-art studies is presented in Table 6. It illustrates a comparison of works using the FD004 C-MAPSS dataset and summarizes the researcher with the publication date, network algorithm, predicted RMSE, and S-score results from previous studies. The value in brackets denotes the standard deviation of the metrics for the experiments in the study.

**Table 6.** Results comparison of different approaches using the FD004 dataset.

| Authors | Approach | RMSE | S-score |
|---|---|---|---|
| Sateesh Babu et al., 2016 [47] | Semisupervised setup | 22.66 | 2840 |
| Zheng et al., 2017 [29] | Deep LSTM | 28.17 | 5550 |
| Listou Ellefsen et al., 2019 [28] | GA + LSTM | 22.66 | 2840 |
| Zhang et al., 2020 [27] | LSTM-fusion | 21.97 | 7726.9 |
| Zhao et al., 2020 [12] | BiLSTM | 24.86 | 5430 |
| Jiang et al., 2020 [26] | BiLSTM-fusion | 29.16 | 7886 |
| de Oliveira da Costa et al., 2020 [25] | DANN-LSTM | 21.30 | 2904 |
| Palazuelos et al., 2020 [31] | CapsNet | 18.96 (0.27) | 2625.64 (266.83) |
| Chen et al., 2020 [32] | Hybrid LSTM with attention | 27.08 | 5649.14 |
| Wang et al., 2022 [23] | B-LSTM | 16.24 | 5220 |
| Qin et al., 2022 [33] | SD-TemCapsNet | 16.49 | 804.05 |
| Ren et al., 2022 [35] | Lightweight and Adaptive KD | 15.10 | 1508.84 |
| Li et al., 2023 [36] | Advertial neurons network | 26.64 (0.29) | 10,343 (783) |
| Proposed method | LSTM-based feature clustering | 16.14 (0.96) | 1299.19 (255.25) |

Accuracy has been improved by using advanced deep learning algorithms, with a deeper network layer or by using fusion algorithms. Compared with work carried out in recent years, the RUL predictions of turbofan engines show the effectiveness of our approach. As some methods tend to have a lower S-score or RMSE, the proposed method achieves better results in terms of the trade-off between performance and explainability, while using a simple architecture with only a single-layer LSTM. For a PHM solution, the more insightful the model, the greater its reliability.

*6.5. Interpretability of Results*

SHapely Additive exPlanation (SHAP) values are a great tool to understand complex tree-based and deep network model outputs. SHAP values can link local and global interpretations. Deep neural networks, known as black-box, are difficult to understand. However, they can be explained by SHAP values, which determine the factors responsible for the outcome and decision. SHAP values are used on independent synthetic features, preventing its limits, in order to understand the role of each factor in an aircraft's turbofan engine.

The results of the average SHAP values of the predictive model are shown in Figure 16. It shows the clusters with the greatest influence on the RUL prediction, placing the largest cluster on top. The x-axis represents the average absolute SHAP values of each cluster. Clusters with larger absolute SHAP values correspond to the most important clusters. Cluster 1 represents the features that contribute the most to the prediction, followed by cluster 5, cluster 3, cluster 2, and cluster 7.

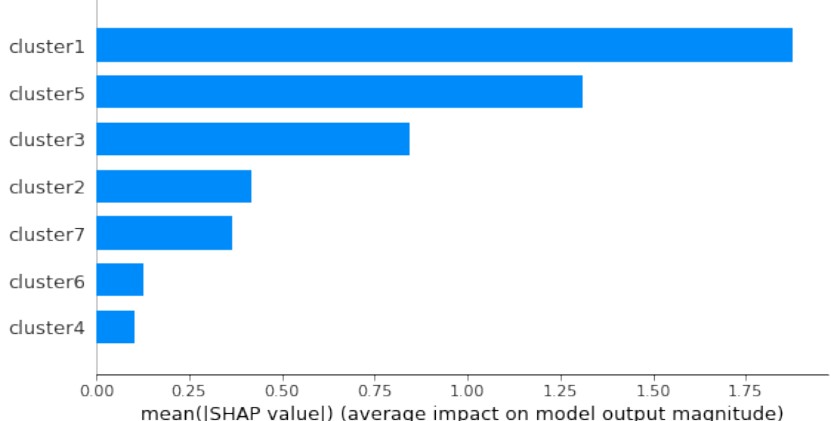

**Figure 16.** SHAP values importance ranking of clusters.

Based on Figure 17, the x-axis represents the SHAP values, and the y-axis represents all clusters. Red color means a high value of a cluster; blue means a lower value of a cluster. Clusters are presented in order of importance of all clusters, with the first cluster being the most important and the last being the least important. This distribution shows the overall impact of the cluster directions. Clusters 1 and 7, with high values, contributed positively to the prediction, while with low values, they contribute negatively. On the other hand, clusters 5 and 3 have a negative influence on the prediction when they have high values, and a positive influence when they have low values.

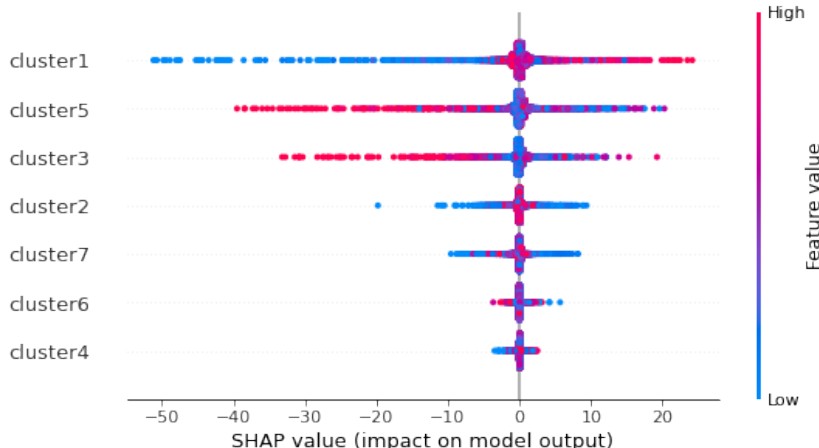

**Figure 17.** SHAP values for the proposed model.

Cluster 2 has a low contribution to the prediction. Clusters 6 and 4 have almost no contribution to the prediction, whether their values are high or low. To relate these results to the initial features, we use the equations found by ClustOfVar in Section 5.2.3. Cluster 1, represented in Equation (10) by the combination of features T24, T30, T50, PS30, and htBleed, has a positive contribution when its values are high, and a strong negative contribution when its values are low. A high value of cluster 1 (for each additional value of these features, cluster 1 decreases by 9.6) corresponds to a low value of total LPC outlet temperature T24, total HPC outlet temperature T30, total LPT outlet temperature T50, static pressure at HPC outlet PS30, and purge enthalpy htBleed. They have the greatest power to predict failures; their higher values increase the remaining useful life and subsequently decrease failure degradation. Cluster 5 has a strong negative contribution when its values are high and a positive contribution when its values are low. A high value of Cluster 5 corresponds to a high value of the physical core speed Nc and a high value of the corrected core speed NRc. That means that a low core speed increases the remaining useful life and so the failure degradation decreases. Cluster 3 with low values does not contribute to prediction, but high values cannot specify contribution to the prediction. This cluster is represented in Equation (12) by the ratio of fuel flow to Ps30 and the total pressure at the HPC outlet. Cluster 2 is represented in Equation (11) by total bypass-duct. It has a low contribution to the prediction. A high value of total bypass-duct does not contribute to RUL prediction. On the other hand, a high value cannot confirm the way of contribution to the prediction. Cluster 7 is represented in Equation (16) by bypass ratio, BPR, negatively, and strongly positively by HPT coolant bleed, W31, and LPT coolant bleed, W32. Low values of HPT coolant bleed and LPT coolant bleed correspond to decreased RUL and thus increased failure degradation. We can conclude that:

1.  The higher the values of total LPC outlet temperature T24, total HPC outlet temperature T30, total LPT outlet temperature T50, static pressure at HPC outlet PS30, and purge enthalpy htBleed, the higher chance of failure due to degradation.
2.  Higher values of physical core speed Nc and a high value of the corrected core speed NRc may result in lower RUL values and, therefore, a higher chance of failure due to degradation.
3.  Higher values of $\phi$, the ratio of fuel flow to static pressure at the HPC outlet, and P30, the total pressure at the HPC outlet, may result in a lower RUL values, resulting in a higher chance of failure due to degradation.
4.  Low values of HPT coolant bleed, W31, and LPT coolant bleed, W32, correspond to decreased RUL and thus increased failure degradation.

## 7. Conclusions

In this paper, we proposed a deep-learning-based feature clustering with an XAI approach to predictive maintenance. The proposed model applied feature clustering on sensors during the preprocessing phase and built one single hidden LSTM layer algorithm. Our model was implemented and evaluated to predict the RUL of engines on the NASA turbofan engine dataset FD004. An ablation study was performed and compared to those obtained without any dimension reduction using LSTM with different numbers of hidden layers and those using a PCA dimension reduction algorithm. The experimental results indicate the effectiveness and suitability of the proposed method. SHAP values were used to explain decisions based on the predictive model outputs, help understand the decision of the LSTM model, and highlight the most important features contributing to the degradation of turbofan engines.

With regard to future work, a promising direction could be followed by applying the technique to other datasets to validate the proposed approach. In addition, another exciting topic is driving causality in post hoc XAI tools to provide more reliable and interpretable explanations and a better understanding of which part of the machine needs maintenance. It provides valuable information to perform proactive maintenance before the aircraft engine fails.

**Author Contributions:** G.Y.: Conceptualization, methodology, formal analysis, validation, writing—review and editing, supervision; A.A.: Methodology, validation, review and editing. All authors have read and agreed to the published version of the manuscript.

**Funding:** This research received no external funding.

**Institutional Review Board Statement:** Not applicable.

**Data Availability Statement:** The C-MAPSS dataset [48] used in this study is publicity available in NASA Ames' Prognostics Center of Excellence (https://www.nasa.gov/content/prognostics-center-of-excellence-data-set-repository/) (accessed on 17 May 2023). The original introductory article is cited as a reference [45].

**Conflicts of Interest:** The authors declare no conflict of interest.

## Abbreviations

The following abbreviations are used in this manuscript:

| | |
|---|---|
| XAI | Explainable Artificial Intelligence |
| PHM | Prognosis and health management |
| LSTM | Long short-term neural networks |
| PCA | Principal components analysis |
| RUL | Remaining useful life |
| C-MAPSS | Commercial Modular Aero-Propulsion System Simulation |
| LPC | Low-pressure compressor |
| HPC | High-pressure compressor |
| HPT | High-pressure turbine |
| LPT | Low-pressure turbine |
| ClustOfVar | Clustering of variables |
| SHAP | SHapley Additive Explanations |
| TW | Time window size |
| RMSE | Root mean square error |
| S-score | Scoring function |

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
