# Peer review of "An Explainable Artificial Intelligence Approach for Remaining Useful Life Prediction"

_aerospace, doi:10.3390/aerospace10050474_

Round 1

Reviewer 1 Report

1. What is "XAI" in the keywords? You need to define it first before abbreviated. Explainable Artificial Intelligence (XAI).

2. Line 170: a hidded state h and a state c should be italics as similar to the equation.

3. Introduction need to be improved by cited reviews related to the type of prognostics i.e. model-based approaches, data-drive approaches, statistical approaches, etc.

4. The fontsize of Figures 4, 6, and is too small. Please revise it.

5. Please also check other Figures with small font size.

6. Data-driven or AI prognostic approaches required training and testing process/stage. Please indicate in the Figure the training stage and the testing stage. Authors could add some information on Figure 13 or other related Figures.

7. Are the methods presented in Table 6 used the similar datasets?

Author Response

Regards.

Reviewer 2 Report

From the point of innovation and framework of the paper, this paper may be considered if the following comments are taken care:

1. About 70% of the article consists of the Introduction, background, and methodology sections with proposed models. I would strongly recommend authors to reorganize as the proposed model can be a separate section and then to explain the experimental analysis with the Results Discussion section of moderate length. 

2. References can also be improved. The authors should thoroughly review the latest research progresses in recent three years. Recent related works could be mentioned to further enhance the reference. Some of the references suits to the proposed work can be

https://doi.org/10.1016/j.asoc.2023.110253

3. The advantages and disadvantages of the previous work are not clearly expounded, in other words, the motivation of writing the paper is not explained. 

4. Various feature clustering and preprocessing techniques were explained as preliminary work. But what was the main purpose of using these methods to satisfy what purpose of the proposed model was not clearly explained? 

5. What was the result of a combination of the LSTM model with Shapely, Why this combination was considered?

6. The concluding lines from 493, are the details which can be extracted due to experience, in such cases how the XAI techniques help to explain the prediction process.

The manuscript should be thoroughly checked for spelling and grammatical mistakes

Author Response

Best regards.

Round 2

Reviewer 1 Report

The paper has been revised according to the reviewer's comments.

Reviewer 2 Report

The authors have given reasonable responses to the comments. The paper can be published in this format. But the Figure1. The structure of LSTM can be removed, as much of the work has described the structure already. Rest all are fine.

Even though the authors have mentioned revising the paper for grammatical checks, It is noticeable to see many grammatical mistakes. Kindly overview again before final submission.